# Molecular and Cellular Insights: A Focus on Glycans and the HNK1 Epitope in Autism Spectrum Disorder

**DOI:** 10.3390/ijms242015139

**Published:** 2023-10-13

**Authors:** Camille M. Hours, Sophie Gil, Pierre Gressens

**Affiliations:** 1INSERM 1141, NeuroDiderot, Neuroprotection of the Developing Brain, Université Paris Cité, 75019 Paris, France; pierre.gressens@inserm.fr; 2Service de Psychiatrie de l’Enfant et de l’Adolescent, APHP, Hôpital Robert Debré, 75019 Paris, France; 3INSERM 1144, Therapeutics in Neuropsychopharmacology, Université Paris Cité, 75019 Paris, France; sophie.gil@u-paris.fr; 4Neurologie Pédiatrique, APHP, Hôpital Robert Debré, 75019 Paris, France

**Keywords:** Autism Spectrum Disorder, placenta–brain axis, glycome, glycoproteins, gene expression, immunomodulation, trophoblast, pregnancy disorders, Slit/Robo

## Abstract

Autism Spectrum Disorder (ASD) is a synaptic disorder with a GABA/glutamate imbalance in the perineuronal nets and structural abnormalities such as increased dendritic spines and decreased long distance connections. Specific pregnancy disorders significantly increase the risk for an ASD phenotype such as preeclampsia, preterm birth, hypoxia phenomena, and spontaneous miscarriages. They are associated with defects in the glycosylation-immune placental processes implicated in neurogenesis. Some glycans epitopes expressed in the placenta, and specifically in the extra-villous trophoblast also have predominant functions in dendritic process and synapse function. Among these, the most important are CD57 or HNK1, CD22, CD24, CD33 and CD45. They modulate the innate immune cells at the maternal–fetal interface and they promote foeto-maternal tolerance. There are many glycan-based pathways of immunosuppression. N-glycosylation pathway dysregulation has been found to be associated with autoimmune-like phenotypes and maternal-autoantibody-related (MAR) autism have been found to be associated with central, systemic and peripheric autoimmune processes. Essential molecular pathways associated with the glycan-epitopes expression have been found to be specifically dysregulated in ASD, notably the Slit/Robo, Wnt, and mTOR/RAGE signaling pathways. These modifications have important effects on major transcriptional pathways with important genetic expression consequences. These modifications lead to defects in neuronal progenitors and in the nervous system’s implementation specifically, with further molecular defects in the GABA/glutamate system. Glycosylation placental processes are crucial effectors for proper maternofetal immunity and endocrine/paracrine pathways formation. Glycans/ galectins expression regulate immunity and neurulation processes with a direct link with gene expression. These need to be clearly elucidated in ASD pathophysiology.

## 1. Introduction

Immunity is a major regulator of the maternal–fetal interaction during pregnancy, with significant neurodevelopmental consequences. The glycome is a major regulator of the immune system [1]. All cells are heavily covered with glycans, which show tremendous diversity and provide important biological information. Glycosylation is an enzymatic process that occurs in the endoplasmic reticulum/golgi apparatus of all cells, with the coordinated action of various glycosyltransferases and glycosidases leading to the attachment of glycans (carbohydrates) to proteins and lipids. It is epigenetically regulated and among the most fundamental post-translational modifications in cellular biology [2].

The development and function of the immune system relies on the dynamic regulation of the expression of glycan-structures and glycan-binding proteins and the interactions between them. In particular, dysregulation of the N-glycosylation pathway has been shown to be associated with autoimmune-like phenotypes. Autoantigens contain a high number of glycoantigens due to numerous N-glycosylation sites and, thus, the role of N-glycosylation can be studied from the perspective of neo-autoantigens [2].

The glycosylation pathway is highly regulated during development, with the expression of sulfated glycoconjugates and, in particular, the carbohydrate HNK1 epitope. Expression of the HNK1 antigen during development is finely regulated, as it takes part in critical functions, such as cell–matrix interactions, cell–cell adhesion, neural-glia cell adhesion, spatial learning, memory, and synaptic plasticity, among others. The human natural killer-1 (HNK-1) epitope is a unique sulfated trisaccharide sequence presented on O- and N-glycans of various glycoproteins and glycolipids. HNK1 is present on O-mannosyl glycans, which are abundantly expressed in the brain. Its characteristic structural feature is a 3-sulfo-glucuronyl residue attached to lactosamine structures (Gal beta1-4GlcNAc) on glycoproteins and glycolipids. There are different HNK-1-specific antibodies described, anti-HNK-1/N-CAM (CD57), Cat-315 and M6749 (according to their affinity to the GlcA residues and sulfate groups of glycans HNK1), with essential functions of the sulfate group of HNK1 responsible for antibody recognition [3]. Glycans and galectins (β-galactoside-binding lectins) play a highly important role in neuronal fate [4]. Indeed, glycans and glycan-related proteins are involved in the determination, growth, and regeneration of axons, synaptic activity, myelin formation and glial cells expression. They also have huge functions in the functioning of the placenta, such as trophoblast infiltration, immunomodulation and angiogenesis [5].

Aside from its expression in cells derived from hematopoietic tissues, HNK1 is expressed in numerous cell populations in the central nervous system, such as glial cells and neuroectoderm and neuroendocrine populations. The presence of HNK1 modifies the N-glycans of several neural cell adhesion molecules of the immunoglobulin super family, such as transmembrane glycoproteins (N-CAM, L1, Ng-CAM, Nr-CAM, TAG-1, F11/13), myelin-associated glycoproteins (MAG, OMgP, SAG, MOG, PMP-22), glutamate receptor GluA2, CD24, extracellular matrix proteins (J1, Integrins, ependymal cells, tenascin-R), phosphacan, and receptor protein-tyrosine phosphatase (RPTP). HNK1 has a central function in neurodevelopment via perineural networks [6,7] and is specifically present on parvalbumin inhibitory interneurons [8]. Hence, the HNK1 epitope is involved in several central and peripheral nervous system processes, including in the hippocampus and cerebellum [9]. Absence of the glucuronyltransferase GlcAT-P or HNK1 sulfotransferase (HNK1-ST) gene results in impaired synaptic plasticity and spatial learning in mice, despite normal birth and brain morphology [10]. These cerebral abnormalities significantly overlap with those found in Autism Spectrum Disorders, which show structural abnormalities, such as an increased number of dendritic spines and decreased number of long-distance connections. HNK1 also shows critical expression in the placenta, exclusively in the extravillous trophoblast, which is the tissue in close interaction with maternal uNK cells and the maternal immune system. The expression of HNK1 and its regulation can be distinctly assessed as part of the brain–placenta axis. Extravillous trophoblast cells express HLA-C, HLA-G, and HLA-E class 1 molecules, which interact with KIRs, CD94/NKG2C, and LILRB1 on uNK cells to prevent immune activation.

We assume that the maternal immune activation responsible for neurodevelopmental disorders is associated with defects in maternofetal interactions based on the interaction between the KIRS on uNK cells at the extravillous trophoblast site and the glycosylated-HLA and the glycan code in the placenta due to immune and sialylated carbohydrate receptor defects, with consequences for epigenetics and gene expression.

This study aims to highlight some molecular and genetic mechanisms associated with glycosylation and glycan/galectin expression in the placenta–brain axis, with a specific focus on the glycan epitope, HNK1, in the ASD pathogenesis.

First, we see that epigenetic modifiers (such as modifications of histones or altered miRNA expression) can alter transcription and impact neurodevelopmental signaling pathways notably Slit/Robo pathway associated with HNK1 expression. Some already known mechanisms of genetic regulation during neurulation are therefore mentioned but few are known, especially in the immune–genetic regulation. Thus a new area of research involving glycan expression is cited as a promising field, GlycoRNA. Secondly, we link the previous cited mechanisms with the defects seen at the maternofetal interface during pregnancy disorders associated with the ASD phenotype and we find the utmost importance of Slit/Robo, HNK1, and other glycan connected pathways in placental functions. We determine that glycan epitopes play a major part in the regulation of the immune barrier and we present the materno-fetal consequences of dysregulation. Finaly we point out the various sites of HNK1 expression, notably in the whole nervous system and other organ function involved in ASD.

## 2. Insights into Transcription Patterns and Gene Expression in Normal and Pathological Neurodevelopment

Research on the placenta–brain axis focuses on the specific field of “the placenta epigenome-brain axis” and the direct impact of placental epigenomic and transcriptomic responses on neurodevelopment. Notably, placental mRNAs linked to inflammatory and apoptotic processes are regulated through miRNAs and correlate with defects in cognitive functions. Defects in placental function notably through placental gene expression and miRNAs expression may directly impair neurodevelopment [11]. However, data are needed on miRNAs tissue-specific differential expression in the placenta and in the nervous system.

Non-coding RNA and aberrant epigenetic profiles, such as DNA methylation and microRNA alterations have been recently identified as the main features of ASD genetics and also considered as potential biomarkers [12]. We infere that transcriptional dysregulation in ASD has neuronal consequences, with the involvement of immune and glycan functions.

### 2.1. miRNAs

miRNAs post-transcriptionally regulate gene expression and have critical functions in the brain, such as in neuronal plasticity and neuronal development [13]. Indeed their interactions with mRNAs in neurons during synaptic stimulation regulate the translation of dendritic transcripts leading to synaptic plasticity. miRNAs are major regulators of protein synthesis at synapses with an active modulation of neuronal signaling processes. Each miRNA has multiple targets and miRNAs collaborate for a specific neuronal response [13]. They can act through NMDA-mediated AMPAR (α-Amino-3-hydroxy-5-methyl-4-isoxazolepropionic acid receptors) expression and trafficking. Other non-coding miRNAs have been identified in synaptic stimulations such as lncRNA or circRNA, but few are known [13]. For example, the study by Vaccaro et al. highlighted alterations in the expression of seven miRNAs in ASD patients: upregulation of miR34c-5p, miR92a-2-5p, miR-145-5p, and miR199a-5p and downregulation of miR27a-3p, miR19-b-1-5p, and miR193a-5p. These mi-RNAs are specifically linked to immunological developmental, the immune response, and protein transcription. Some affect the SIRT1 (synaptic plasticity and memory formation), HDAC2 (an immediate early genes belonging to histone deacetylase enzymes), and PI3K/Akt-TSC:mTOR signaling pathways. miR-145 targets establishment of the neural crest through Sox9 [14]. Moreover, the study by Noroozi et al. on the miRNA-mRNA network contributing to ASD pathogenesis showed the implication of five different modules: neurexins and neuroligins, glutamatergic synapse, cell adhesion molecules, NOTCH, MECP2 and circadian clock pathways, L1CAM interactions, and neurotransmitter release cycle [15].

### 2.2. Slit/Robo Signaling and HNK1

ASD is associated with Slit/Robo signaling defects during axon guidance, dendritic spine formation, progenitor cell proliferation, migration, and neocortical formation [16]. miRNA can directly modulate Slit/Robo signaling and heparan sulfate biosynthetic enzymes, seen in vascular organization during development [17]. Slit is a secreted glycoprotein and a ligand of the Robo receptor. Two Slit/Robo-mediated signal transduction systems are involved in formation of the cerebral cortex: one involving the Slit/Robo GTPase-activating protein (srGAP) and the other cadherins. Slit/Robo binding results in downstream signaling, leading to actin depolymerization, axon repulsion, and the collapse of dendritic spines. Cadherins interact with adaptor proteins, such as catenin, to contact the actin cytoskeleton. Cadherin-catenin interactions up or downregulate adhesion. Slit/Robo binding results in beta-catenin phosphorylation, its translocation to the nucleus, and transcriptional activation [16].

Glycan epitopes, such as HNK1, are involved in the Slit/Robo signaling-mediated proliferation and differentiation of neural tube cells. The Slit/Robo1 pathway controls neuroepithelial cell proliferation and differentiation in a time-dependent manner. Dorsoventral neural tube genes are finely spatiotemporally regulated. According to a study by Wang et al., Robo1 has a substantial effect on several pathways notably for the Sonic hedgehog (SHH) expression in the ventral neural tube and the resulting production of migrating neural crest cells that express HNK1. Anti-HNK1 antibodies specifically inhibit the attachment of cranial neural crest cells to laminin but not collagens I or IV or fibronectin, whereas they inhibit the attachment of trunk neural crest cells to collagens I and IV [18]. Components of the extracellular matrix and other signaling molecules play an important role in the dorsoventral migration of HNK1-expressing neural crest cells [19]. In addition, HNK1 carbohydrate epitopes on glycoproteins are expressed on heparan sulfate [20] and it has been demonstrated that heparan sulfate directly modulates the Slit/Robo interaction. Without heparan sulfate, there is no repulsive activity of Slit, consequently resulting in excessive dendritic spine formation [21] (Figure 1).

Slit can inhibit the cell cycle by blocking Wnt signaling. It is important to note that protein N-glycosylation in the Wnt/β-catenin signaling pathway and E-cadherin-mediated cell-cell adhesion have a common conserved regulatory molecule, β-catenin. β-catenin can act as a structural ligand of E-cadherin junctions, as well as a co-transcriptional activator of the Wnt/β-catenin signaling pathway, which targets the N-glycosylation-regulating gene, DPAGT1 [22]. The DPAGT1 gene is encoded on chromosome 11q23.3, which is highly associated with the risk of ASD, notably for the KMT2A gene. It encodes a transcriptional coactivator with histone H3 lysine 4 (H3K4) methyltransferase activity that mediates chromatin modifications associated with epigenetic transcriptional activation. It has an important function in regulating gene expression during early development via H3K4 [22]. We will see that several mechanisms are involved in gene regulation and may explain the regulatory effects of genes on each other.

### 2.3. Mecanisms of Genetic Expression Regulation Implicated in Normal and Pathological Neurodevelopment

ASD is a genetic disorder and complex genetic regulation mechanisms occur from the earliest stages of development, in response to the immune environment. Many mechanisms have yet to be understood, but some are known and may explain genetic data in ASD. The following information has been extracted from the work of D. Duboule (2020) on the regulation of developmental genes, demonstrating that the gene positioning is consistent with the development of structures, notably in neurulation. It is of note, that euchromatic loci and gene regulatory elements, such as enhancer sequences, are highly involved in ASD and explain the molecular phenotype [23]. Thus, the POGZ gene, which codes for a chromatin regulator, the Pogo Transposable Element with ZNF Domain protein, has deleterious genetic variants that are very strongly associated with ASD. The POGZ gene shows genomic linkage in the developing forebrain at euchromatic loci and gene regulatory elements [23]. Regulatory mutations affect multifunctional transcription factors and signaling pathways. Structural mutations induce syndromic pleiotropic effects. Chromosomal conformation capture studies have shown specific and privileged interactions of different areas of chromatin linked with the genetic diversity of expression in ASD. Indeed, disconnection between a gene and enhancers in the nervous system causes inactivation of the gene in the central nervous system (CNS) and leads to specific pathological phenotypes. Similarly, a gene can carry regulatory sequences of another nearby gene within its introns. Thus, deletion of a specific gene can give rise to a specific phenotype, but not necessarily the converse, which means a specific phenotype (such as that of ASD) can be linked to diverse genetic modifications. Duplications, microdeletions, and single-base mutations can lead to a common phenotype, notably by acting on the intron. Similarly, activating or repressing transcription factors can act at various levels, explaining the diversity for the same phenotype. Areas of strong chromatin cohesion, loop formation, cohesin involvement, and chromatin binding areas explain the important effect of genomic topology on transcriptional regulation and the temporal and spatial expression of functional genes. It is o note, that HNK1 and sulfoglucuronic carbohydrate antigens belong to multigene families, which leads to diverse consequences of genetic modification.

### 2.4. GlycoRNA

A new field of research concerning specific conserved, small, non-coding RNA-glycan conjugates has recently arisen: that of glycoRNAs. They are highly enriched with sialic acid and fucose. They are present on the cell surface, can be modified, with complex N-glycan types, and can be recognized by Siglec receptors and anti-RNA antibodies [24]. This finding represents a tremendous advance in the field of cellular interactions. It is the first step towards understanding a chemical pathway of the utmost importance: the immune–genetic pathway, through Siglecs and antibodies potentially involved in ASD pathogenesis. Specific immune patterns in the tolerance of implantation occur in ASD pregnancies, acting on the genetics and further neurodevelopment of the embryo through glycosylation. Therefore, this could be an interesting field of research into the complex genetic of ASD.

## 3. Insights into the Maternofetal Interface: HNK1 and Placental Glycosylation Processes

It has been demonstrated placental pathology predicts the onset of neurodevelopmental disorders [25].

### 3.1. Slit/Robo in the Uterus and Placenta

Trophoblast-decidual interactions underlie common diseases of pregnancy, including preeclampsia and preterm birth. Other functions are relevant, such as the expression of Slit/Robo in the uterus, with different physiological and pathological functions in the reproductive and fetal systems. There may also be an influence of hormonal and endocrine factors on Slit/Robo expression in reproductive tissues.

The Slit/Robo pathway has a regulatory function in angiogenesis and the functions of VEGF/FGF2 in the placenta. Indeed, Slit/Robo binding keeps vascular integrity through the inhibition of abnormal angiogenesis and endothelial hyperpermeability. It plays a central role in directional migration and the formation of vascular networks. Therefore Slit/Robo pathway has an autocrine and paracrine function in angiogenesis and trophoblast functions [26]. It is specifically involved in pregnancy disorders with endothelial dysfunction, such as preeclampsia [26]. Slit2 and Robo1 are expressed in the syncytiotrophoblast and Slit3, Robo1, and Robo4 in the capillary endothelium of the placental villi. In preeclamptic conditions, levels of Robo1, Robo4, and sFLT1(soluble fms-like tyrosine kinase-1) increase significantly. Hypoxia dysregulates the expression of some Slit and Robo proteins. Placental functions may be regulated by an autocrine/paracrine pathway, with co-expression of Slit/Robo by the trophoblast and endothelium [27]. The effects of neurotoxins identified to be causal factors for neurodevelopmental disorders were studied on DNA methylation in the placenta and fetal brain in mice. In both, differentially methylated regions were enriched in neurodevelopment, cellular signaling functions, and regions of bivalent chromatin (characterized by the dual presence of histone modifications H3K4me3 and H3K27me3, associated with transcriptionally active and repressed chromatin). These regions shared the same subset of genes enriched for Wnt signaling, Slit/Robo signaling, genes differentially expressed in neurodevelopmental disorders. The regions identified in this study overlap with differentially methylated regions found in human neurodevelopmental disorders in the brain and placenta [28].

At the same time, a study by Tiensuu et al. showed involvement of the Slit/Robo signaling pathway in spontaneous preterm birth. Moreover, Slit/Robo signaling in trophoblasts affects the expression of immune-response-modifying genes, such as inflammatory cytokines and chemokines. By affecting the actin cytoskeleton, dysfunction of the Slit/Robo pathway may have a direct effect on the villous organization of the placenta and its diverse functions, such as exosome trafficking, fibrinoid deposits, and the extracellular matrix [29,30]. Slit has a fundamental anti-inflammatory effect in the human placenta. Specifically, inflammatory mediators associated with specific pathological conditions downregulate Slit and enhance placental inflammation. As a consequence, Slit has a crucial function in fetal growth and neurodevelopment [31].

Dysregulation of heparan sulfate and HNK1 expression in the placenta may have a direct effect on the Slit/Robo pathway. This leads to a pathological state, such as preeclampsia, hypoxia, or preterm birth, which are the specific pathological pregnancy conditions associated with ASD. Further studies are needed to understand the role of HNK1 in placental homeostasis.

In parallel, a study by Liao et al. found the mean expression of placental SHH protein and mRNA in early and late-onset preeclampsia were higher than that of women of a late gestational age control group. There were no differences in the mean serum SHH levels between the early and late preeclampsia groups of the study. At the same time, placental mitochondria oxidative stress injury and apoptosis of trophoblast cells were higher in the preeclampsia group than in the late gestational age control group. In addition, the addition of SHH serum to trophoblast cells reduced oxidative stress versus the addition of a preeclamptic serum [32]. During preeclampsia, SHH has a direct effect on the oxidative stress mechanism in trophoblast cells. SHH and HNK1 expression evolved in a parallel manner [14] and may be associated with the level of oxidative stress in the preeclamptic state. Abnormal Wnt/β-catenin signaling pathway is described in preeclampsia but some studies are needed for details in the pathophysiology [33].

### 3.2. Involvement of Other Glycan Epitopes That Bind to the Siglec Family: CD22, CD24, CD33, and CD45

Trophoblast glycans are essential for placental function and development. They also modulate innate immune cells at the maternal–fetal interface and promote maternofetal tolerance. As a consequence, immune dysregulation at the maternofetal interface may influence the functions of trophoblast glycans which may, in turn, activate an immunogenic response. In addition, glycosylation and glycan trophoblast dysregulation may alter maternofetal immune tolerance, angiogenesis, and neurodevelopment.

There are numerous glycan-based pathways of immunosuppression. One engages the CD22 inhibitory signal in antigen-specific follicular B cells. B cells present trophoblast antigens only to CD4 T cells. Thus, B-cell suppression is associated with the loss of cognate CD4 T cells. Trophoblast antigen glycans likely modulate CD22-Siglec interactions via the recruitment of endogenous sialylated proteins [34]. The CD22 protein is a member of the immunoglobulin superfamily and is homologous to three cell adhesion proteins: carcinoembryonic antigen, myelin-associated glycoprotein, and neural cell adhesion molecule [35].

CD24 and CD33 are also highly expressed in villous and extravillous cytotrophoblasts. CD24 is also present in maternal uterine glands. Their interactions with siglecs may have a mediating effect on the immune tolerance at the maternofetal interface [36]. A study by Bleckmann et al. on the mouse brain showed that CD24 and the cell adhesion molecule L1 interact by O-linked glycans carrying α2,3-linked sialic acid. They observed the expression of mucin-type and O-mannosyl glycans carrying functional epitopes, such as 3-linked sialic acid, disialyl motifs, LeX, sialyl-LeX, and HNK1 units. The study provides evidence of the contribution of carbohydrate determinants to CD24-mediated biological activities [37]. CD33 belongs to the immunoglobulin (Ig) superfamily and its siglec cognates deliver negative signals via cytosolic tyrosine-based regulatory motifs [38]. A study of myeloid-derived suppressor cells (MDSCs) within the decidua showed the presence of specific CD33 populations (CD33/HLA^-^DR^-/-^ and /HLA^-^DR^+/-^) expressing iNOS, IDO, and a specific cytokine profile that may be an important regulator of immune tolerance [39].

CD 45 is a receptor protein tyrosine phosphatase and a highly glycosylated cell-surface protein. It initiates and modulates T-cell receptor signaling and its glycosylation changes with the life of the T cell. This glycosylation state modulates intracellular signaling by the cytoplasmic tyrosine phosphatase domain and the response of the T cell to antigens [40]. CD45 glycosylation regulates the T-cell response to antigens, activation, and immune function, such as cytokine production [39]. Galectin-1 is also a regulator of T-cell cytokine production, downregulation of Th1 cytokines, and upregulation of Th2 cytokines. This effect may occur with the same glycan ligand regulating Th1/Th2 susceptibility. Galectin 1 is highly expressed in the thymus and binds to N- and O-glycans on different glycoprotein receptors [41]. CD45 defects lead to T and B cell dysfunction, with an important consequence in autoimmune diseases [42]. ASD children exhibit significant dysregulation of CD45 expression, suggesting that CD45 has a function in the immune abnormalities and altered neurodevelopment observed in ASD [43].

### 3.3. Autoimmunity and Antibodies at the Maternofetal Interface

The maternofetal interface is regulated by immunological interactions between fetal trophoblasts and maternal leucocytes bearing receptors for HLA molecules. HLA genes are involved in immune dysfunction and autoimmune diseases. Genetic polymorphism in the HLA region is associated with the ASD phenotype [44]. The HLA genes exhibit a high degree of polymorphism and epitope sharing between a large or small number of HLA molecules is thus described. It is the shared epitopes that explain the multitude of possible immunizations following an encounter with a single antigen, defined as serological cross-reactivity groups (CREG cross-reactive groups). Such cross-reactivity is likely what generates autoimmune processes.

Autoimmune activation is present in ASD patients and altered NK cells have also been reported to be part of the ASD phenotype [45]. Moreover, circulating maternal autoantibodies and neuronal dysfunction in neonates has been described in ASD, in particular, involving maternal anti-brain autoantibodies targeting neurodevelopmental proteins [46]. Maternal-autoantibody-related (MAR) autism is related to central, systemic, and peripheral autoimmune processes. Specific autoimmune mechanisms are associated with NK dysregulation [47,48]. It has been shown that innate inflammation leads to NK-cell activation, with the disruption of regulatory T-cell (Treg) function, and that specific genetic variants modulate the cytotoxic activity of NK cells. Autoantibodies to KIR have also been identified in various systemic autoimmune conditions. NK cells have a modulating effect in immune-mediated pathology [49]. The persistence of maternal immune cells in newborns is associated with diverse autoimmune disorders. Indeed, alloreactive maternal T cells can lead to a graft-versus-host-like reaction or alter the development of Tregs in newborns, leading to autoimmunity. These maternal microchimeric T cells can lead to gut inflammation in the offspring and adverse autoimmune events. Such immunogenic mechanisms can occur during maternal immune activation [50].

A study by Jennewein et al. highlighted the preferential selection of antibodies with glycan modifications and their ability to activate NK cells to cross the placenta to the neonate. Fc glycosylation and di-galactosylated Fc-glycans may be crucial in the placental transfer of antibodies to activate innate immune cells [51].

Trophoblasts and endothelial cells produce an IgG with one of its Fab arms asymmetrically glycosylated. The presence of this extra carbohydrate is important for immune protection of the placenta and the fetus, as well as for trophoblast invasion and endothelial development for a healthy pregnancy. Impairment can result in pathological pregnancy conditions associated with ASD susceptibility in mothers, such as spontaneous abortion or preeclampsia [52].

Similarly, glycans are also crucial for maternofetal tolerance through glycan-mediated B-cell suppression. B cells are inhibited by CD22-LYN signaling, with direct suppression by the sialylated glycans of trophoblast antigens. This molecular pathway is all the more important, as B cells directly modulate MHC-class-II restricted presentation of antigens to CD4^+^ T cells, with consequent T-cell suppression. Glycosylation is of the utmost importance in non-immune self-recognition in the placenta and the subsequent prevention of immune-mediated pregnancy complications that may lead to pathological placental structure and function, as seen in ASD pregnancies. Glycans are also involved in the regulation of the antigen immune response to autoimmunity [34]. A study by Ziganshina et al. explored the specificity of immune tolerance in preeclampsia. Placenta-blocking antibodies can prevent the activation of the maternal immune system against fetal allogeneic cells. In preeclampsia, there is a restricted repertoire of placental antibodies relative to a healthy pregnancy that are less variable and qualitatively different from those in the peripheral blood. This is explained by a decrease in anti-glycan antibodies, which are crucial for maternofetal tolerance. Consequently, the mechanisms of chronic rejection observed in preeclampsia are associated with lower levels or the absence of such anti-glycan antibodies, resulting in the exposure of unmasked foreign patterns, leading to the activation of effector functions. Such antibodies also regulate anti-inflammatory and anti-thrombotic reactions and prevent placental and endothelial dysfunction [53].

As a consequence, immune activation at the maternofetal interface (such as neurotoxins, infections, maternal autoimmunity, HLA haplotypes) may lead to immunogenic processes involving cytokines, lymphocytes, NK-specific activation, and the production of antibodies against specific placental and fetal epitopes, with the crucial involvement of placental glycosylation. These modifications have important consequences for transcription and genetic expression. Importantly, the HNK-1 reactive carbohydrate is known to be highly immunogenic. Indeed, monoclonal antibodies against various embryonic nervous system antigens are specifically found against the HNK1 carbohydrate. We can assume that immunogenic reactions may target the HNK1 antigen expression regulation present on the extra-villous trophoblast, the placental tissue directly contacting maternal cells (Figure 2).

Moreover, the HNK1 epitope is associated with early neuronal organization, neural cell migration, and axonal growth patterns. Nearly all neural crest cells express HNK1 reactivity early after the initiation of migration to form the neural tube. They migrate in the embryo along defined pathways and differentiate into neurons, glia, sensory cells, sympathetic and parasympathetic ganglia, neuroendocrine cells, melanocytes, and various other cell types. The HNK1 carbohydrate is involved in the outgrowth of motor neurons on ventral roots and motor nerves. It is expressed by myelinating Schwann cells of the basal laminae of ventral spinal roots and at lower levels by the myelinating Schwann cells of dorsal roots. It is also associated with numerous dendrites and astrocytic processes. HNK1 has crucial functions in the cerebellum, shown in rat studies, notably with its association with Purkinje cells and their dendrites. There is a differential rostro-caudal expression phenotype due to distinct cell-lineages during neurogenesis. Sulfoglucuronyl carbohydrates are involved in cell recognition and adhesion and bind to the extracellular matrix glycoprotein laminin at the heparin binding site. The ligation of sulfoglucuronyl carbohydrates to laminin at critical stages of neural development is an important physiological marker. They also bind to amphoterin which is a non-histone chromosomal protein. The sulfoglucuronyl carbohydrates binding to amphoterin is regulated through its cell surface and nucleic acid binding properties, with a direct impact on the nucleus of the cell and DNA alterations [54]. As a consequence, antibodies against HNK1 proteins and sulfoglucuronyl carbohydrates lead to various types of neurological impairment, with alterations of axons and endothelial cells and macrophage infiltration. Circulating anti-sulfoglucuronyl carbohydrate antibodies in brain microvessels can disrupt the blood-brain barrier or blood-nerve barrier, with the penetration of antibodies to the nervous parenchyma. Anti-HNK1 and sulfoglucuronyl carbohydrate antibodies were shown to inhibit the outgrowth of astrocytic processes and disrupt neural crest migration by disrupting the interaction of neural crest cells with the extracellular matrix and their binding to laminin [55]. Altered connective tissue and extra-cellular matrix due to laminin defects are a common pattern described in ASD phenotype [56].

These modifications lead to defects in neuronal progenitors and the function of the central nervous system, specifically due to molecular defects later in the GABA/glutamate system.

## 4. Insights on HNK1 Expression in Organs

HNK1 is highly expressed in neurodevelopmental pathways (Figure 3) and in various organic functions (Figure 4).

### 4.1. In Eye Formation

Neurodevelopmental disorders, notably ASD, are significantly associated with eye abnormalities [57].

The HNK1 carbohydrate epitope has a fundamental role in every component of the eye [58], with expression both on glial and neuronal cells [59]. HNK1 may participate in neuron-to-neuron and glia-to-neuron adhesion in the retina. One of the phosphacan/RPTP beta isoforms bearing the HNK1 epitope is dually expressed in the retina and neurons [60]. A large number of HNK1-positive cell clusters have been shown to be located around developing eyes in silver mutant quail embryos, with fibroblast growth factor 1 (FGF), laminin, and heparan sulfate proteoglycans as cofactors located in the neural retina, the retinal pigment epithelium, and the choroidal tissues [61]. Similarly, expression of the Rho GTPase Slit-Robo (SRGAP2) has been reported in the optic nerve in a rat model [62]. The dysfunction of HNK1 and Slit/Robo in ASD could be related to the observed pathophysiology of the eyes.

### 4.2. In the Brain

The HNK1 carbohydrate is highly involved in proteoglycans, predominantly expressed early in development, and aggrecan-HNK1 is tightly associated with keratan sulfate chains [63]. The eye and the brain also abundantly express keratan sulfate chains [64]. HNK1- and Cat315-related epitopes have also been found to be expressed in specific immune cells, notably astrocytes, and HNK-1 is a glial marker expressed in Schwann cells [65].

#### 4.2.1. RPTP Signaling

A study by DINO et al. showed the HNK1 epitope (Cat-315) to be expressed in perineuronal net (PNN) structures (RPTP and aggrecan) at each step of synaptogenesis [66]. At the same time, N-acetylglucosaminyl-transferase is highly expressed in the brain. Its activity is required for the O-mannosyl-linked HNK1 modification of the developmentally regulated and neuron-specific receptor protein-tyrosine phosphatase (RPTP). This leads to a decrease in cell-cell adhesion and increased migration on laminin. Such N-acetylglucosaminyl-transferase activity results in higher levels of phosphorylated catenin, which alters cell adhesion. RPTP glycosylation increases galectin-1 binding and the retention of RPTP on the cell surface. Therefore, N-acetylglucosaminyl-transferase regulates RPTP signaling, modifying cell–cell and cell–matrix interactions during neurogenesis [67].

#### 4.2.2. SHANK and Ankyrin Protein

A study of Ishioka et al. showed the presence of an identical amino acid sequence for the N-terminal portion and peptides between the ankyrin-binding protein of rat brain, a member of the immunoglobulin superfamily, and the HNK1 molecule purified from soluble fractions of bovine brain [68]. HNK1 may be secreted from an extracellular site of ankyrin-binding protein of the SHANK proteins involved in the glutamate receptor and the cytoskeleton related to ASD pathophysiology.

ASD mouse models present both brain-region specific alterations in cell-surface glutamate receptors and a decrease in the density of cell-surface GABA receptors in these brain regions [69]. This suggests that mutations in genes believed to be involved in ASD are not sufficient to explain the molecular phenotype, as there is no common molecular phenotype for the surface expression of glutamatergic and GABAergic receptor subunits. It may rather be due to the specific molecular regulation of glutamate and GABAergic receptors on the cell surface.

#### 4.2.3. NMDA Signaling

Shared regulation of expression is found between HNK1 (Cat315 epitope) glutamatergic and GABAergic axon terminals in parvalbumin-positive neurons and the NMDA receptor [70]. Indeed, memantine, which is a blocker of the NMDA receptor, downregulates the HNK1 epitope in the hippocampus. Moreover, there is a specific link between different ASD-risk alleles (coding neuroligin, Shank2, TBR1, GluD1 for example) and reduced function of the NMDA receptor and/or abnormal switch activation between GluN2B and GluN2A subtypes [71]. In parallel, Mascio et al. demonstrated the importance of the regulation between the density of Cat315 (HNK1) PNNs in the barrel cortex and sensory deprivation. Indeed, mGlu5, involved in the formation of the somatosensory map in the barrel cortex, regulates PNN formation according to sensory stimulation. Thus, mGlu5 receptors directly regulate the shaping of PNNs in response to sensory stimulation. They have a primary function in the developmental plasticity of parvalbumin interneurons. This study showed the major function of the mGlu5/PNN axis in ASD and the link between HNK1 expression and the regulation of sensory stimulation [72].

#### 4.2.4. Slit/Robo Signaling

As seen above, the abnormal structure of ASD brains, showing increased dendritic density and a reduced number of long-distance connections, is likely to be associated with deficiencies in the expression of heparan sulfate containing HNK1 and mediation of the Slit/Robo interaction [21].

#### 4.2.5. Astrocytes

Yamada et al. reported the detection of HNK1 in astrocytes [70]. A link was also reported between a deficiency of the N-acetylglucosaminyltransferase-IX that synthesizes HNK1 and the downregulation of astrocyte activity [73].

More specifically, CNS glial cells have been shown to strongly express protein tyrosine phosphatase receptor type zeta (PTPRz) in the membrane, which can be modified by glycosylation, including (O-mannosyl)-HNK1. HNK1-O-Man-PTPRz is increased on the surface of astrocytes involved with demyelination. Thus, there is a direct link between PTPRz glycosylation, astrocytes, and demyelination [74].

A study conducted on maturation of the optic nerve head (ONH) showed that all glial cells in the fetal ONH express HNK1/NCAM. In particular, type 1B astrocytes express HNK1/NCAM and finely regulate communication between blood vessels and other tissues [75]. This also highlights the important function of HNK1 in the formation of the eye.

### 4.3. In Other Organs

HNK1 is also involved in the function of other organs. It has been found in the lung, specifically within intrapulmonary arteries in the endothelium, internal elastic lamina, and smooth muscle layer, where it is a functional marker for the development of the air-blood interface in the fetal stage [76].

In another condition, fetal growth restriction is associated with cardiac dysfunction at birth. It is important to note that fetal growth restriction is most frequently due to uteroplacental insufficiency. It is believed that uteroplacental insufficiency directly hinders cardiac development. The study by Chou et al. concluded that fetal growth restriction is also associated with decreased glycogen deposition and HNK1 content in cardiomyocytes. It is possible that uteroplacental insufficiency, resulting, among other defects, in cardiac deficiency, may also be associated with abnormal HNK1 levels [77].

Two glucuronyltransferases (GlcAT-P and GlcAT-S) and a sulfotransferase regulate HNK1 biosynthesis. In the kidney, the non-sulfated HNK1 carbohydrate is predominantly expressed on the apical membranes of the proximal tubules in the cortex and has also been detected in the thin ascending limb in the inner medulla, as the GlcAT-S mRNA is highly expressed relative to in the brain. Conversely, GlcAT-P and HNK-1 sulfotransferase mRNAs are highly expressed in the brain but not in the kidney [78].

## 5. Conclusions

Glycosylation plays central roles in both the regulation of immunity during pregnancy in maternofetal tolerance and in fetal and postnatal neurodevelopment. It also interacts with cytoskeletal regulation and gene expression. Small changes in glycosylation during pregnancy can greatly alter functions essential for a healthy pregnancy and proper neurodevelopment [5]. Glycan epitopes, such as the HNK1 epitope, are also of primary importance in the glutamatergic and GABAergic circuits of parvalbumin neurons in PNNs and are also expressed in extravillous trophoblasts in direct contact with maternal tissue. Moreover, HNK1 is directly involved in the regulation of oxidative processes and the initiation of neurite outgrowth and neuronal development, with coordinated regulation by Amphoterin and RAGE [54]. We need to clearly elucidate glycan-regulated immune mechanisms that occur during the early stages of pregnancy for fetal tolerance under the conditions of ASD. Inflammation can directly induce changes in N-glycosylation, with consequences on immune cell processes, such as pathogen recognition and the regulation of inflammatory responses. Indeed, proinflammatory cytokines can affect the expression of glycosyltransferases, allowing the biosynthesis of N-glycans. The consequences concern both innate and adaptive immunity and result in modifications of effector functions, cell interactions, and signal transduction [79].

## Figures and Tables

**Figure 1 ijms-24-15139-f001:**
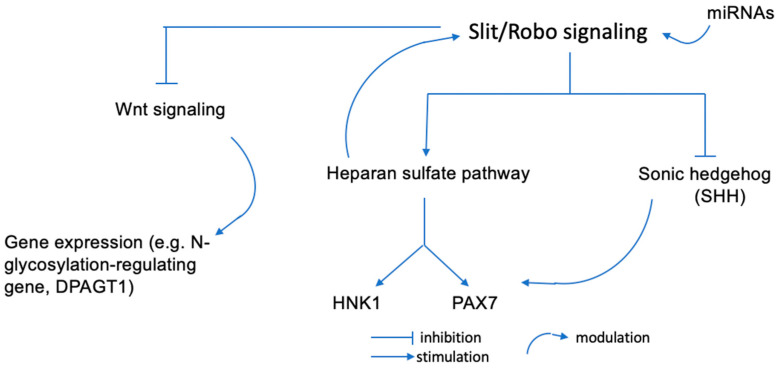
Potential regulation pathways of cranial neural crest cells production. miRNAs can influence Slit/Robo binding and functions. Slit/Robo can impact the expression pattern of the dorso-ventral neural tube genes. Heparan sulfate moduiates the Slit/Robo interactions and SHH can establish a ventro-dorsal gradient in the neural tube to regulate the PAX/HNK1 family gene expression.

**Figure 2 ijms-24-15139-f002:**
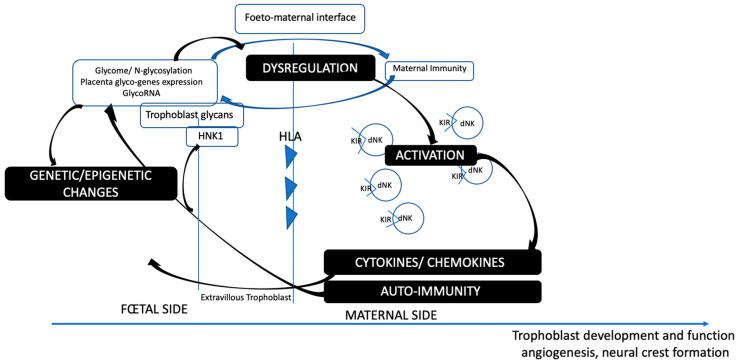
Glycome-immune regulation at the foeto-maternal interface.

**Figure 3 ijms-24-15139-f003:**
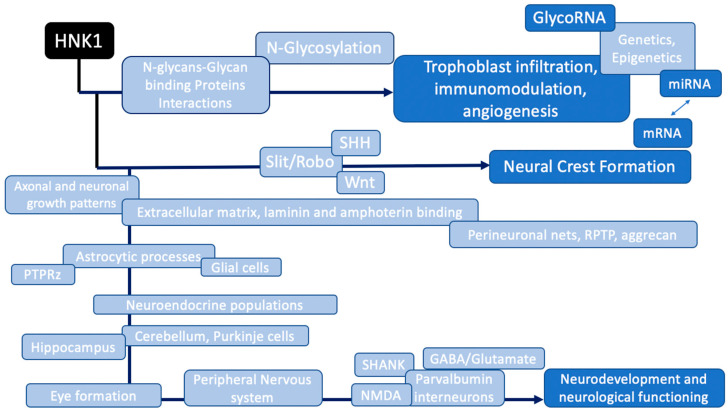
HNK1-regulated pathways in neurodevelopment linked to Autism Spectrum Disorders.

**Figure 4 ijms-24-15139-f004:**
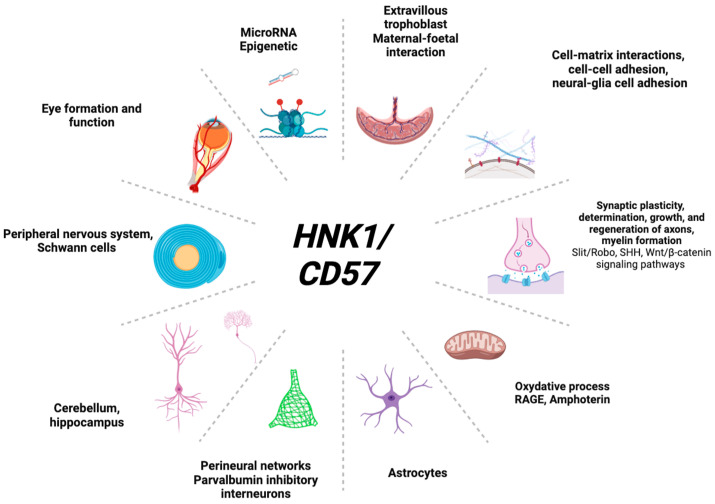
HNK1-relevant organic functions.

## Data Availability

Hyperlinks and persistent identifiers for the data are given and freely available.

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
