# Peer review of "Molecular and Cellular Insights: A Focus on Glycans and the HNK1 Epitope in Autism Spectrum Disorder"

_ijms, 2023, doi:10.3390/ijms242015139_

Round 1

Reviewer 1 Report

The perspective entitled "Molecular and cellular insights: A focus on glycans and the HNK1 epitope in Autism Spectrum Disorder" included the spin abnormalities reported in the Autism. The glycosylation-immune placental processes further implemented in the neurogenesis have been described. Further, the function of glycan epitopes viz., CD57, CD22, CD24, CD33 and CD45 have been described. 

Minor comments:

1.   The effect of miRNA on the neuronal plasticity may be explained more in the section 1.1.1.

2. Slit/Robo signaling is reported to be important in Autism. The mechanism should be explained with the help of illustration. 

NA

Author Response

I would like to thank you very much for your valuable review of this work.

As you recommended, I have added additional explanations for a better understanding of the effects of miRNAs on synaptic plasticity. For the second point the precise mechanism underlying slit/robo mediated corticogenesis need still to be clarify but the article highlights the specific implication of HNK1 in the Slit/robo functioning and the link in the ASD pathogenesis. I have added an illustration explaining the links between slit/Robo, miRNA, SHH and HNK1 that may be involved in the ASD pathogenesis.

Reviewer 2 Report

This perspective aims to summarize the expression and function of glycans and the HNK1 epitope in Autism Spectrum Disorder (ASD). ASD is considered a complex neurological disorder primarily driven by genetic mutations, with some cases potentially influenced by environmental factors, such as toxins and adverse prenatal conditions.

In light of the article's title, the author endeavors to explore the relationship between glycans and the HNK1 epitope in the context of ASD, offering a promising starting point. However, the article suffers from a disjointed structure, excessive off-topic discussions, and a prevalence of statements and paragraphs lacking adequate citations or robust references to substantiate the author's claims. Some descriptions of conclusions or results in certain references are overly generalized, contributing to the article's poor readability. Here are some specific instances:

In the abstract section, the author repeatedly uses "they" three times without clear reference to the preceding subject (from lines 25 to 28). Additionally, sentence structures are convoluted, resulting in confusion and difficulty in comprehension.

In the introduction section, the sentence from lines 72 to 74 lacks proper references. The statement from lines 83 to 84 fails to clarify the functions of galectins in placenta and brain development and lacks appropriate references. In the final paragraph, the author claims the study's aim is to highlight the critical role of glycosylation, glycan, and galectin expression in the placenta-brain axis, with specific focus on the HNK1 glycan epitope. However, the discussion of glycosylation and glycan expression in the placenta-brain axis is limited. The sentence from lines 92 to 93 is particularly perplexing.

The sentence from lines 97 to 99 is confusing, and the provided reference does not support the author's assertions. It is puzzling why this reference is labeled as number 75 yet appears early in the article.

The paragraph spanning lines 113 to 120 is loosely connected to the study's topic and lacks appropriate references. More references should be integrated to substantiate the author's speculations.

The sentence from lines 145 to 146 is unclear.

The sentence from lines 161 to 162 is puzzling.

The sentence from lines 165 to 167 lacks references.

The paragraph from lines 168 to 188 is off-topic and lacks references.

The sentence from lines 241 to 242 is overly general.

The sentences from lines 272 to 276 lack references.

The paragraph from lines 350 to 366 lacks references.

Numerous other issues exist that cannot be exhaustively listed here. In summary, this article necessitates significant restructuring and revision prior to publication.

The English language should be improved to avoid confusing sentences.

Author Response

I thank you very much for your review. 

Your comments primarly concern the structure of the article. I have taken them into account to improve the structure of the article : I have modified the organisation of the chapters, I have added a paragraph at the end of the introduction to make it easier to understand how the thought process unfolded, and I have re-worked, removed or added some information and illustration, following the comments and the recommandations you give.

For some of your comments I am not sure I perfectly understand what you mean by "disjointed structure, excessive off-topic discussions, lack of adequate citations or robust references, overly generalized" since all the information in this article has been thought out and presented in a precise order so that the reader can make the link between the information. The article is structured in such a way that the elements given refer to each other in the different parts and are interconnected to each other. For example mecanistic informations can be given in one section and they may explain the link made between the different data in another section. It is important to note that scientific data is still missing to complete the reflexion, so I added phrases to expain it. This may explain the feeling of vagueness or over-generalization you are talking about. But all the information has been carefully selected, given in a precise order and referenced. The idea of this article is to investigate the possible links in the molecular and biochemical mechanisms involved in ASD, highlighting the central role played by glycans both at the materno-foetal interface and in neurodevelopment, and targeting an epitope of interest, but with data in the litterature still lacking in the area. The idea is to consider new avenues of study.

Concerning each precise point you underline, I thank you very much, I made the modifications requested.

Reviewer 3 Report

Where are the legends for the figures?  Why do the words in the figures all have the red line indicating misspelling under them?  Where is all the information regarding contributions, funding ,etc.

Editing is needed throughout the paper with regards to word choice, run on sentences and grammar.  For example, in some cases it is Slit-Robo and in others it is Slit/Robo.  The abstract in particular needs editing for clarity.  Line 206 - 'Slot' vs. 'Slit'.

Author Response

I would like to thank you very much for your proofreading and for the corrections you have identified. I have made all the modifications requested.

Reviewer 4 Report

The manuscript "Molecular and cellular insights: A focus on glycans and the HNK1 epitope in Autism Spectrum Disorder, written by Hours C, Gil S and Gressens P. in a review presenting the possible roles of glycans and of HNK1, specifically modified and glycosylated epitope in development of autism spectrum disorders. The manuscript is presented under the subtitle Perspective.

In the Introduction, the importance of the glycosylated proteins and expression of the HNK1 epitope in the brain and immune cells are presented. In more details Slit/Robo signaling in the brain development is described. In the second part, the role of Slit/Robo pathway, and glycosylated proteins on maternofetal interface during pregnancy are presented, and in the third part expression of HNK1 in different organs and roles in brain function are described.

The manuscript could be better organized, more focused on the topic and connected.  There are lots of data, but also superficial phrases and data not directly connected with the matter. At the same time, I miss some common and general introduction on glycosylation in the immune system, hypotheses on autism development and better explanation of the autism, as well the basic mechanisms by which they could be connected. It could be explained that autism could be linked to early influence of placenta and maternofetal interface, but also to some other causes. Many genes, signaling pathways and molecules are mentioned, but more explanations are needed, especially when they are mentioned for the first time.

Other comments:

when presenting some idioms for the first time, it could be explained, as well as its abbreviation (line 51)

sentence reorganization: line 63, 191

line 106: where are miRNAs expressed?

line 130: how can beta catenin affinity be lowered?

line 139: dorsoventral

line 144: Slug cells?

line 164: what is the connection between KMT2A and DPAGT1 gene (except that they are on the same chromosome)?

lines 180-188: better explanation

lines 204-209; what is bivalent chromatin

lines 234-239; what does is mean SHH serum and that SHH and HNK expression have evolved in parallel manner?

some references are cited twice in the References, and they are not correctly ordered in the text.

when presenting HLA genes it is said that they are involved in immune dysfunction (this is not their primary role) and that "Genetic polymorphism in the HLA region is associated with the ASD phenotype" "The high polymorphism is responsible for sharing epitopes...?" (lines 288-290)

line 319, line 478: there could be better distinction between authors' hypothesis and data from literature

line 366: references missing

Figures have too small fonts

English language is fine

Author Response

I thank you very much for your proofreading and your comments.

Concerning the structure of the article, it is an important element and one that led to the majority of your comments : I thought the inital chapter numbering did not make it possible to see clearly how elements relate to each other so I have changed the organisation of chapters, I have added a paragraph at the end of the introduction to make it easier to understand how the thought process unfolded, and I've reworked, removed or added some information.

The article has been designed in such a way that the elements given refer to each other in the different parts and are interconnected to each other throughout the article. For example, mecanistic informations can be given in one section and they may explain the link made between the different data in another section, particularly for the question of genetics the "regulation of gene expression" section provides a set of mechanistic elements that occur in gene expression and that are likely to occur in ASD. 

This paper is focused on the analysis of the relevant molecular and biochemical pathways regulated through the glycans/galectins expression and implicated both at the materno-fetal interface and in neurodevelopment. It highlights some possible regulations and it investigates new avenues. Therefore scientific data are missing to complete the reflexion and I have precised it to make it clearer for the reader.

All the information in this article has been thought out and presented in a precise order so that the reader can make the link between the information.

Concerning each precise point you underline, I thank you very much, I made the modifications requested.

Round 2

Reviewer 2 Report

Thank you for authors' reply.The revised manuscript boasts a well-organized structure for describing the pathogenesis of ASD, with particular emphasis ont he role of HNK1 in its etiology. Additionally, the suggested references had been added , rendering it a readable review article.

Author Response

I thank you very much for your valuable review and your constructive advice to make the text clearer.

Reviewer 4 Report

The manuscript "Molecular and cellular insights: A focus on glycans and the HNK1 epitope in Autism Spectrum Disorder, written by Hours C, Gil S and Gressens P. in a review presenting the possible roles of glycans and of HNK1, specifically modified and glycosylated epitope in development of autism spectrum disorders.

The authors made some changes and improved the structure of the manuscript, but still some parts of the manuscript are not enough precise.

The second chapter is titled "Insights into transcription patterns and gene expression" without the object. So, several general processes are described, although sometimes there are no direct connections with autism. It is written that "Non-coding RNA and aberrant epigenetic profiles have been recently identified as the main features of ASD genetics", and that "miRNAs are major regulators of protein synthesis at synapses".  These statements should be better explained and have proofs. Also Figure 1 seems to me oversimplified explanation of regulation pathways, especially considering the role of miRNAs. Some topics are not connected together, such as in lines 170-177: from the gene DPAGT it is jumped to the regulation of histone modifications. Next, talking about gene regulation (again very superficial title), it is said that "gene positioning is consistent with development of structures. Also, following sentences (lines 182-3 and 187-189) need explanation and are too generalized. Describing GlycoRNA, although it is an interesting topic, there are no data and proofs of its association with ASD.

Furthermore, in some articles cited, the topic is neurodevelopment, not specifically autism (like when there was treatment with PCB). When talking about autoimmunity it is written "The high polymorphism of HLA genes is responsible for the sharing of epitopes between a large or small number of HLA molecules. " This paragraph needs additional explanation. When talking about amphoterin, it is not clear whether this molecule is in cytosol or in ECM and how is "their binding to amphoterin regulated through its cell surface and nucleic acid binding properties....".

When writing about HNK in organs it is written that neurodevelopmental disorders, notably ASD are significantly associated with eye abnormalities".  Reference is needed. On line 466 reduced function of NMDA was connected with different ASD risk alleles, but they are not mentioned.

On the other side, the only article connecting NK cells and autism which I have found, showed decreased number of NK cells in the blood of patients with autism. Also, it is not mentioned that HNK1 disorder is found to be linked to neuropathy. Figure 3 has too small text.

I think that it should be discerned what are facts, data from published articles and what are hypotheses, based on possible links.

Author Response

I thank you very much for your careful proofreading to make the text even more precise. In that way I add terms such as "we can assume, suppose" "it is possible" "may" to distinguish more clearly between the assumptions (formulated on the basis of known data) and the data established, which are all referenced in the text.

For the second chapter, I had to set the context of the explanations given for the reader : indeed the informations both concern normal and pathological neurodevelopment, general informations on normal neurodevelopment are required before considering them in a pathological context. 

I add additional informations and reference about epigenetic profils, transcription and ASD. 

The figure 1 is an attempt to summarize the as yet insufficient and unconfirmed data in the literature on the regulation between slit/robo, the heparan sulfate pathway and SHH. This seems to be the maximum valid scientific data possible to give.

The explanation concerning the DPAGT1 gene is not a jump, since the particularity of the mecanistic regulation of genes is then addressed in the following section. I add a phrase to explain it.

I give a more precise title for the 2.3. chapter and I add the source of informations concerning the mechanisms cited.

For glycoRNA I have changed the presentation of this point, so as to simply put forward the hypothesis that it would be potentially interesting to look at it in autism.

I have added the other various precisions and corrections you mentionned.

Studies linked HNK1 with neuropathy such as this one : 

Delmont E, Attarian S, Antoine JC, Paul S, Camdessanché JP, Grapperon AM, Brodovich A, Boucraut J. Relevance of anti-HNK1 antibodies in the management of anti-MAG neuropathies. J Neurol. 2019 Aug;266(8):1973-1979. doi: 10.1007/s00415-019-09367-0. Epub 2019 May 14. PMID: 31089861.

I thank you again for your valuable review.

Round 3

Reviewer 4 Report

The authors responded to my comments and improved the manuscript.

However, I still have some comments.

line 56: better explanation

When I commented the sentence ""miRNAs are major regulators of protein synthesis at synapses" the point was that miRNA regulates protein synthesis, on ribosomes, and ribosomes are in the body of the nerve cell, not on the synapsis.

line 131: few are

Considering lines 177 -185, and the DPAGT1 gene, the point of the comment was that the role of DPAGT1 gene is described, and then it is written that it is on 11th chromosome, which codes for KMT2A gene etc. Beside their loci and possible involvement in neural development, there is no direct connection between KMT2A and DPAGT

line 193: Of note, euchromatic loci and gene regulatory elements, such as enhancer sequences, are highly involved in ASD and 1explain the molecular phenotype.  Enhancer sequences and gene positioning are involved in all developmental processes. Possibly their deregulation can be involved in ASD.

line 195: mutations in regulatory sequences?

line 199: All the processes are present in all cell types, and these descriptions are too generalized. If there are some specific changes (exact genes, loci, transcription factors) characteristic for ASD, they can be mentioned and described in more details.

210: several cytokines are involved in genetic regulation? Which cytokines, where? Cytokines are extracellular molecules, their signaling can influence gene expression, but, again, without details and genes it can stand for any cell.

line 211: also, last sentence has no links with previous text

line 236: The Slit/Robo pathway has a regulatory function in angiogenesis and the functions of VEGF/FGF2 in the placenta. This sentence needs additional explanation.

lines 245-249: In both, differentially methylated regions were enriched in neurodevelopment, cellular signaling functions, and regions of bivalent chromatin (characterized by the dual presence of histone modifications 4me3 and H3K27me3, associated with transcriptionally active and repressed chroma-tin). These regions shared the same subset of genes enriched for Wnt signaling, Slit/Robo signaling, genes differentially expressed in neurodevelopmental disorders.

It is not clear what it means. Differently methylated regions of what? in comparison with what (neurodevelopment as a process normally involves processes of methylation of some genes and expression of other genes). Possibly mentioned processes are categories in pathways analyzed in bioinformatics.

line 326: The HLA genes exhibit a high degree of polymorphism and epitope sharing between a large or small number of HLA molecules is thus described.

lines 400-405: the same comment as previous time: it is not clear how cytosolic protein can be bound to extracellular glycoprotein. The facts should be explained, there is no use of just mentioning something that is not clear or overgeneralized.

line 422: organ function

line 443: astrocytes are not immune cells

line 463: how can a molecule be secreted from an extracellular site of a protein?

line 475: what does it mean shared regulation of expression?

line 481: Mascio et al. demonstrated the importance of the regulation between the density of Cat315 (HNK1) PNNs in the barrel cortex and sensory deprivation.- better explanation

Author Response

I thank you for your comments which helped to clarify specific points in the discussion and to correct some inaccuracies left in the text. 

I have added the details requested. For some points, the explanations followed the introductory sentence of the idea.

For line 400-405 : it is said that ankyrin-binding protein shares an identical amino acid sequence with the HNK1 molecule, and that HNK1 may be secreted from an extra-cellular site of ankyrin-binding protein (probably in exocytosis traffic). The point is to show described link between HNK1 and the SHANK gene (with dysregulated  expression in autism).

Astrocytes participate to immune functions in the brain : they are regulators of innate and adaptative functions in the injured CNS.

Some of the points raised lack scientific data at the present time, which is why this article has been placed under "perspectives" to allow us to present new avenues of research that are still in the pipeline.

Thank you for your very conscientious proofreading.